

# The atmospheric branch of the hydrological cycle over the Indus, Ganges and Brahmaputra River basins

Rogert Sorí[1], Raquel Nieto[1,2], Anita Drumond[1], Sergio M. Vicente-Serrano[3], Luis Gimeno[1]

[1] Environmental Physics Laboratory (EphysLab), Universidade de Vigo, Ourense, 32004, Spain
[2] Department of Atmospheric Sciences, Institute of Astronomy, Geophysics and Atmospheric Sciences, University of SãoPaulo, São Paulo, 05508-090, Brazil
[3] Instituto Pirenaico de Ecología, Consejo Superior de Investigaciones Científicas (IPE-CSIC), Zaragoza, 50059, Spain

*Correspondence to:* Rogert Sorí (rogert.sori@uvigo.es)

**Abstract.** The atmospheric branch of the hydrological cycle over the Indus, Ganges, and Brahmaputra river basins in the South Asian region was investigated. The 3-dimensional model FLEXPART v9.0 was utilized. An important advantage of this model is that it permits the computation of the freshwater budget on air parcels both backward and forward in time trajectories from 0.1 and 1000 hPa in the atmospheric vertical column. The analysis was conducted for the Westerly Precipitation Regime (WPR) (November-April) and the Monsoonal Precipitation Regime (MPR) (May-October) in the period from 1981-2015. The main terrestrial and oceanic climatological moisture sources for the IRB, GRB and BRB and their contribution to precipitation over the basins were identified. For the three basins, the most important moisture sources for precipitation are i) on the continental regions, the land masses to the west of the basins (in this case called West Asia), the Indian region (IR) and the basin itself, and ii) from the ocean, the utmost sources are the Indian Ocean (IO) and the Bay of Bengal (BB), and it is remarkable that despite the amount of moisture reaching the IRB and GRB from land sources, the moisture supply from the IO seems to be first associated with the rapid increase/decrease in precipitation over the sources in the MPR. The technique of the composites was used to analyse how the moisture uptake spatially vary from the sources (the budget of evaporation minus precipitation *(E-P)* was computed in a backward experiment from the basins) but during the preonset and predemise dates of the monsoonal rainfall over each basin; this confirmed that over the last days of the monsoon at the basins, the moisture uptake areas decrease in the IO. The Indian region, the Indian Ocean and the basins itself are the main sources of moisture responsible for negative (positive) anomalies of moisture contribution to the basins during composites of driest (wettest) WPR and MPR.

## 1 Introduction

Research of the hydrological cycle in the Asian region has been extensive, which is mainly because of the strong influence of the Asian Summer Monsoon (ASM), which develops a crucial role in moisture transport and the supply of precipitation in this region (Webster, 2006). The ASM system has three different but interrelated components: South Asian (SAM), Southeast Asian (SEAM) and East Asian Monsoon (EAM) (Janowiak and Xie, 2003). The Indian Summer Monsoon (ISM) is one of the most studied phenomena and is part of the SAM. It develops in response to the large thermal gradients between the warm





Asian continent to the north and the cooler Indian Ocean to the south (Slingo, 1999). Solar heating is considered a fundamental driver of all of the monsoon systems. Heating of the Tibetan Plateau leads to increased ISM rainfall via enhancement of the cross-equatorial circulation and a concurrent strengthening of both the Somali Jet and westerly winds that bring moisture to southern India (Rajagopalan and Molnar, 2013). Surface heating over the plateau plays a role in producing cyclonic vorticity

in the shallow lower-layer but negative vorticity in the deep upper layers through atmospheric thermal adaptation (Yanai and Wu, 2006; Song et al., 2010). The satellite and conventional observations support an alternative hypothesis, which considers the monsoon as a manifestation of seasonal migration of the intertropical convergence zone (ITCZ) (Gadgil, 2003). Understanding and predicting the variability of the Indian monsoon is extremely important for the well-being of over one billion people and the diverse flora and fauna inhabiting the region (Gadgil, 2003).

The monsoonal regimes in India, tropical Africa, and North America are provided with moisture from a large number of regions (Gimeno et al., 2012). According to Misra et al. (2012), instead of rainfall, evaporative sources (of the ISM) may be a more appropriate metric to observe the relationship between the seasonal monsoon strength and intra-seasonal activity. It is worth mentioning that the precipitation over any area of land comes from the moisture already available in the local atmosphere, the convergence of the moisture advected into the region by the winds, and the supply by evaporation from within

the same region (Gong and Eltahir, 1996; Trenberth, 1999). The atmospheric branch of the hydrological cycle consists of the atmospheric transport of water, which is mainly in the vapour phase (Peixoto and Oort, 1991), and plays a crucial role in understanding the bridge between evaporation in the sources and precipitation over remote regions. Indeed, the identification of moisture sources for precipitation constitutes an important feature to understand the further mechanisms associated with rainfall variability (Gimeno et al., 2012) and it has become a major research tool in the analysis of extreme events (e.g., floods

and droughts) (Gimeno, 2014).

Numerous studies (e.g., Drumond et al., 2011; Misra and DiNapoli 2014; Ordoñez et al., 2012; Pathak et al., 2017) have determined the origin of moisture that contributes to precipitation in Asia. Ordoñez et al. (2012) confirmed the key action of the Somali Low-level Jet bringing moisture from the Arabian Sea and the Indian Ocean during the boreal summer and documented the importance of recycling as the main water vapour source in the winter for this region. Chen et al. (2012)

identified and quantified the origin (destination) of moisture and air mass transported to (from) the Tibetan Plateau from June to August, and Pathak et al. (2017) made an extensive study of the role of oceanic and land moisture sources during the summer monsoon in India to confirm the strong land-ocean-atmosphere interactions. To determine the evaporative sources of the Southeast Asian Summer Monsoon (SEAM) region, Misra and DiNapoli (2014) found that the largest evaporative source for the rainy season in the SEAM region came from the local land-based evaporation and the seas in the immediate vicinity.

Tuinenberg et al. (2012) applied a water trajectory model to investigate the moisture recycling rates over the Ganges River basin (GRB) and confirmed that a large influx of moisture from the Indian Ocean dominates precipitation. The recycling of precipitation helps in defining the role of land-atmosphere interactions in the regional climate (Bisselink and Dolman, 2008).





The Indus River basin (IRB) is located in the northwest of India. Utilizing stable isotope measurements, Karim and Veizer (2002) determined the predominant moisture sources for the IRB were located in a closed basin such as the Mediterranean or other inland seas. Together, the IRB, the GRB, and the Brahmaputra River basin (BRB) are the largest Asian river basins and occupy a large part of the Indo Gangetic plain. In these basins, the importance of the basin itself in providing moisture has

been previously proven (COLA, 2017).

Nevertheless, due to the complex hydrological cycle over the Indo-Gangetic, this region is quite unique compared to the rest of the world and the ASM plays a crucial role. In this region, the moisture source identification and evaluating their role and moisture contribution are fundamental for understanding the nature of the precipitation. For these reasons, the aim of this work was to investigate the atmospheric branch of the hydrological cycle over the Indus, Ganges, and Brahmaputra river basins.

This was done first by identifying the main seasonal oceanic and terrestrial moisture sources for each basin and later quantifying their contribution to precipitation over the basins. This analysis will allow determination of the role of the sources during different precipitation regimes, specifically for the rainfall associated with the monsoon onset and demise and for dry and wet conditions over the basins. Different criteria have been used in the past to define the onset and retreat over different monsoon regions and even over different parts of the same monsoon (Zeng and Lu, 2004). Taniguchi and Koike (2006) argue

that the rapid enhancement of the wind speed related well with the abrupt beginning of the rainy season and it represents a clear transition in atmospheric conditions or the beginning of ISM.

## 1.1 Study Area

The study was performed for the Indus, Ganges and the Brahmaputra river basins, which are located in South and Southeast Asia (Fig. 1). The Ganges is the largest river in the Indian subcontinent followed by the IRB and the BRB; all of these river

basins are densely populated and represent a complete range of landscapes and ecosystems on which the major agricultural activities rely (Davis, 2003; Hossen, 2015; Tare et al., 2015; Mahanta et al., 2014, Shaw et al., 2011).

Two main climate systems drive the annual precipitation over the basins, the Asian Summer Monsoon during the winter months and the Western Disturbances (WD) (Hasson et al., 2014) provided that some feature a bimodal precipitation regime: the Monsoonal Precipitation regime (MPR) for May-October and the Westerly Precipitation regime (WPR) for November-April

(Hasson et al., 2016; Hasson et al., 2014). In the MPR, the summer monsoon has a key role in the Hydro-climatology of Asia. Even the subseasonal river discharge is found to be strongly tied to the monsoon intraseasonal cycle, which results in a near-in phase timing of the Ganges and Brahmaputra discharge (Jian et al., 2009), whereas the WD during the WPR are important synoptic weather systems responsible for almost one third of the annual precipitation over the northern Indian region and most of the cold season precipitation (Dimri el al., 2015). During the boreal winter, the meltwater is extremely important in the

Indus basin and is also important for the Brahmaputra basin but plays only a modest role for the Ganges (Immerzeel et al., 2010). Indeed, the IRB irrigation System (IBIS) is the largest irrigation system in the world (Qureshi, 2011; Laghari et al.,



2012). From a geographic and climatologic perspective, the IRB is at a transition between the monsoon system in the east and the Mediterranean one in the west (Karin and Veizer, 2002).

## 2 Materials and Methods

### 2.1 The Lagrangian approach

The 3-d Lagrangian model FLEXPART v9.0, which was developed by Stohl and James (2004, 2005), was utilized to identify the moisture sources for the IRB, GRB, and BRB and investigate their role in the atmospheric water balance over the basins. The model was executed considering the atmosphere is homogeneously divided into approximately 2.0 million uniformly distributed parcels. The parcels were advected backward and forward in time utilizing the 3-dimensional (3D) winds field from the ERA-Interim Reanalysis (Dee et al., 2011), which is a mechanism described by equation 1:

$$dx/dt = v[x(t)] \tag{1}$$

where x is the position of the parcel and *v [x(t)]* is the wind speed interpolated in space and time. For each parcel, a constant mass (m) was considered. By interpolating *q* to *x(t)*, the net rate of change of the water vapour content of a particle is computed by equation 2, where e represents the moisture gain (through evaporation from the environment) and *p* the moisture loss (through precipitation).

$$(e - p) = m\,(dq/dt) \tag{2}$$

Integrating *(e - p)* in all of the atmospheric vertical column, we obtain a diagnosis of the surface freshwater flux, which is represented by (E-P) (Stohl and James, 2004) in equation 3 and where *K* is the number of particles residing over an area *A*.

$$E - P \approx \frac{\sum_{k=1}^{k}(e-p)}{A} \tag{3}$$

To calculate the freshwater flux, it was considered as the average time residence of the water vapour in the atmosphere, and it
was set to 10 days according to Eltahir and Bras (1996) and Numaguti (1999). The calculus conducted in the air masses tracked backward in time from over each basin and permitted identification of those regions where air masses gained and lost humidity before arriving at the basins, and thus, enabled the identification of the moisture sources of the regions. This indicates that those regions where *(E-P)>0* are considered moisture sources, whereas the opposite *(E-P)<0* are moisture sinks. FLEXPART needs five three-dimensional fields: horizontal and vertical wind components, temperature, and specific humidity in the
ECMWF vertical hybrid coordinate system. The model also needs the two-dimensional fields: surface pressure, total cloud cover, 10 m horizontal wind components, 2 m temperature and dew point temperature, large-scale and convective precipitation, sensible heat flux, east/west and north/south surface stress, topography, land-sea-mask, and subgrid standard deviation of the





topography. To run FLEXPART, it utilized the ERA-Interim reanalysis data set (Dee et al., 2011) at 6 h intervals (00, 06, 12, and 18 UTC) and at a resolution of 1 ° in latitude and longitude considering 61 vertical levels from 0.1 to 1000. Detailed information regarding FLEXPART functionalities can be found in Stohl and James (2004; 2005). Concerning the limitations of the method, the equation (3) can diagnose *(E-P)* but not *E* or *P* individually according to Stohl and James (2004). These

authors also argue that along with individual trajectories, q fluctuations can occur for nonphysical reasons (e.g., because of q interpolation or trajectory errors), which is a limitation that is partly compensated among the many particles in an atmospheric column over the target area.

This approach has been used in numerous studies with the main purpose of characterizing the atmospheric branch of the hydrological cycle in different regions, e.g., in Western and Southern India (Ordoñez et al., 2012), the Sahel (Nieto et al.,

2006), China (Drumond et al., 2011; Huang and Cui, 2015), the Mississippi River Basin (Sthol and James 2005), the Amazon River basin (Drumond et al., 2014), and Central America (Durán-Quesada et al., 2010). At the global scale, FLEXPART has been implemented to identify the main oceanic and continental moisture sources for precipitation (Gimeno et al., 2012) and a catalogue of moisture sources for two sets of continental climatic regions (Castillo et al., 2014). The main advantage of FLEXPART is that it permits the tracking air masses backward and forward in time and calculated along trajectories of the

water balance in the atmospheric column.

For delimiting the most evaporative regions in the moisture sources, some authors (e.g., Drumond et al. 2014, 2016a) have used a threshold (a percentile value) to define the boundaries. In this work, we apply the same technique; the value of the 90th percentile in the *(E-P)>0* values integrated over 10 days of transport was considered to delimit the sources. An exception in this work was that each river basin was considered a source region; which permitted the study of the role of each one and the

balance of *(E-P)* on them. Once delimited, the moisture sources were implemented in a forward in time analysis to determine the contribution of each source to the precipitation over the basins (when *(E-P)i10<0*). This analysis allowed us to later perform a seasonal correlation analysis between the data of *(E-P)i10<0* with precipitation and potential evapotranspiration to determine the best linear relationships.

### 2.2 Selection of preonset and predemise monsoonal dates over the basins

Here, we address the spatial variability of the moisture uptake for the basins during composites of dates associated with the pre-onset and demise of the Indian monsoon. To determine the day on which the increase in rainfall indicates the beginning of the monsoon involvement for each basin, was utilized the method proposed by Noska and Misra, 2016. This method is based





on daily cumulative anomalies (*C'm*) of the average precipitation for each basin (*P*) along the year and according to equations 4 and 5.

$$C'm(i) = \sum_{n=1}^{i}[Dm(n) - C],$$ (4)

$$C = \frac{1}{MN}\sum_{m=1}^{M}\sum_{n=1}^{N}D(m,n)$$ (5)

*D (m,n)* is the daily basin rainfall for day n of year m, and *C* is the climatology of the annual mean of the precipitation at each basin over *N* (=365/366) days for *M* years. The onset is then defined as the day after *C'm* reaches its absolute minimum value but from May onward when the MPR is defined. When applied, this criterion avoids the selection of a false date that could arise and be associated with the previous winter precipitation. Similarly, the demise is considered the day when *C'm* reaches the maximum value after the onset. For this analysis, it was necessary to use a series of precipitation on a daily basis over an

extended period of the study, and in this case, data was available for 1981 – 2015 from the Climate Hazards Group InfraRed Precipitation with Station data CHIRPS (Chris et al., 2015).

To identify dry and wet conditions in the basins, the Standardised Precipitation-Evapotranspiration Index (SPEI) (Vicente-Serrano, et al., 2010) was used. SPEI is based on a standardization of the climatic water balance (Precipitation –P- minus Atmospheric Evaporative Demand –AED), which is computed on different time-scales. The data of P and AED was obtained

from CRU TS v.3.24.01. We calculated the 6-month SPEI to assess drought severity conditions over the three basins since this time-scale adapts to the time period of the two main rainfall seasons over the basins (WPR and MPR). Thus, the 6-month SPEI at the end of November (October) characterized the soil-water balance for the WPR (MPR). According to the criterion of Mckee et al. (1993), we used an SPEI threshold of +/ -1.5 to identify severe and extreme drought and wet conditions.

Different criteria have been used in the past to define onset and retreat over different monsoon regions and even over different

parts of the same monsoon (Zeng and Lu, 2004). Taniguchi and Koike (2006) argue that the rapid enhancement of the wind speed related well with the abrupt beginning of the rainy season and it represents a clear transition in atmospheric conditions or the beginning of ISM. To determine the onset and demise dates, we applied an objective index to the basins from Noska and Misra (2016), which was previously adapted for the Asian Monsoon region in Misra and DiNapoli (2014) and builds upon the index proposed by Liebmann et al. (2007). The analysis is based on the cumulative anomalies of daily rainfall averaged

(see equations 4 and 5) over the basins and is permitted to identify the date associated with rainfall increase because of the monsoon onset (the day after the minimum accumulated rainfall anomalies) and demise (the day of the maximum accumulated



rainfall). According to Noska and Misra (2016), this index is capable of representing the annual rainfall variability across the region and thus must be adequate for our target regions.

## 2.3 Selection of the seasonal period and the data utilized

The study was conducted for the period from 1981-2015 and taking into account the criterion of Hasson et al. (2016). These authors considered two hydro-climatological periods of the year: May-October, which was named as "Monsoonal precipitation regime" and hereafter (MPR) and November–April, the "Westerly precipitation regime", which was hereafter (WPR) to study the seasonal cycle of the water balance over the Indus, Ganges, Brahmaputra and the Mekong River basin. The data of the Vertical Integrated northward and eastward Moisture Flux belongs to the ERA-Interim Reanalysis (Dee et al., 2011) with a resolution of 1 ° × 1 ° and the data of the precipitation and potential evapotranspiration were obtained from the CRU TS v.3.24.01 (Harris et al., 2014) with a spatial resolution of 0.5 °. The concept of the potential evapotranspiration (PET) has proven to be inappropriate because the evaporation climatic demand is not only linked to the climate but also to the type of the evaporative surface, and some authors have adopted a more suitable term: evaporative atmospheric demand (AED) (Katerji and Rana, 2011; McVicar et al., 2012). Hence, several authors utilize the more suitable AED.

## 3 Results and discussion

### 3.1 The water balance over the basins

The mean annual cycle of the Precipitation (P) and Potential evapotranspiration (Pet) over the Indus, Ganges and Brahmaputra basins appears in Figure 2. For the three basins, the maximum Pre occurs during the summer months, whereas the maximum Pet occurs in April and May. The annual cycle of Pre from CRU data are very similar to that of Hasson et al., (2014), who analysed the seasonality of the hydrological cycle over the same basins for the 20th century climate (1961–2000 period) utilizing PCMDI/CMIP3 general circulation models (GCMs) and observed precipitation data. In IRB, the Pre annual cycle is characterized by two maximum peaks in March and July-August (Fig. 2a). In this basin, the Pet remains higher than the precipitation across the year; in fact, Cheema (2012) argue that the major part of this basin is dry and located in arid to semiarid climatic zones. Laghari et al. (2012) also found for the climatology from 1950–2000, the Potential evapotranspiration at the IRB exceeds the precipitation over it across the year. The Pet annual cycle differs from that obtained by Hasson et al. (2014) and these authors utilized the surface upward latent heat fluxes from all of the models they used to compute an ensemble evaporation dataset. In the GRB, the annual cycle of Pet is very similar with that obtained for the IRB, but the precipitation is higher and reaches the maximum rate (10 mm/day), which is also in July-August (Fig. 2b). From June to October, the Pre exceeds the Pet. Over the BRB, the monthly average precipitation rate increases abruptly from March until a maximum (~11.5 mm/day) in July and later falls until a minimum is reached in December (Fig. 2c). The Pet does not surpass 4 mm/day in the annual climatology and it less than that obtained for the IRB and GRB. The main meteorological situations that cause heavy





rainstorms over the Brahmaputra basin are due to the shifting of the eastern end of the seasonal monsoon trough to the foothills of Himalayas in the north and the 'Break' monsoon situations during the monsoon season (Dhar and Nandargi, 2000). These results suggest the BRB is wetter than the western GRB and IRB; this is because the monsoon rainfall dominates in the summer months in the eastern region and gets weaker on the western side with a time delay of a period of weeks (Hasson et al., 2014).

### 3.2 Identification of moisture sources

The climatological budget of *(E-P)i10* obtained in the backward track experiment of air masses residing over the three basins and the Vertically Integrated Moisture Flux (VIMF) and its Divergence appears in Figure 3. The analysis was conducted for the WPR and MPR periods. In the first one, the most intense positive values (delimited by p90) in the pattern of *(E-P)i10*

obtained for the IRB, are over the basin itself and they extend southwest until the Indian Ocean (IO) and East Africa (EA). High *(E-P)i10 >0* values are also confined by the p90 (green line) to the west of the basin (hereafter West Asia; WA), the Persian Gulf (PG), the Red Sea (RS) and to the southeast occupying a major part of the Indian region (IR) and part of the Bay of Bengal (BB). In this season, the field of *(E-P)i10* obtained in the backward experiment from the GRB is very similar to the one obtained for the IRB, but the p90 is now extended to the east and even confines part of the Sea of China (CHS). Over the

GRG itself, the highest values of *(E-P)i10 >0* are observed. For the BRB in the pattern of *(E-P)i10*, the line of p90 is longitudinally extended from East Africa until the CHS and seems less intense than those previously obtained for the IRB and GRB. In this season, the prevalence of the divergence of the VIMF can be distinguished in almost all of the Indian regions except the northern parts of the IRB and the GRB and the western parts of the BRB, where they are overcome by the convergence of the VIMF. In the northern part of the basins, the VIMF is mainly to the east but over the Indian region is mainly

to the southwest and is more intense over the Arabian Sea, which is a feature that well known to be linked to excessive latent heat fluxes and both related to the anomalous meridional temperature gradient originated between the lands to the north of the Arabian Sea and elsewhere and the Sea Surface Temperature (SST) at the Arabian Sea (Levine and Turner, 2012; Marathayil et al., 2013).

In the MPR, the pattern of *(E-P)i10* is more extended and intense than in the WPR (Fig. 3). In the backward experiment for

the three basins it is commonly distinguished that the p90 line comprises a huge area in the western Indian Ocean and to the west of each basin. The moisture transport from the Indian Ocean crossing the Arabian Sea and penetrating into the continent is revealed by the VIMF; observational analysis shows strong monsoons depend on moisture fluxes across the Arabian Sea (Levine and Turner, 2012). According to Qiao et al. (2013), the interannual variation of the moisture source over the western-central South Indian Ocean is determined by the variation of both local precipitation and evaporation. Thus, the use of

FLEXPART to assess the role of this region in moisture supply to the target regions could be an advantage. Previous regions that provided moisture for the basins in the East Asian region and the China Sea are moisture sinks in this season in accordance with the VIMF convergence. To the east of the IRB, over the east of the GRB and over all of the BRB are moisture sinks. In these areas, the air masses lose humidity before they arrive at each basin, which is apparently because of the intense precipitation over this region associated with the Indian Monsoon.



To determine the different roles within the continental and oceanic moisture sources and taking into account the region where they are located, we made a separation for the WPR and MPR. The selected sources are shown through a schematic representation in Figure 4. The regions shaded in colour represent the location and spatial extension of the most important moisture sources previously delimited using the p90 values and independently calculated for the *(E-P)i10>0* values for every

basin and period (Fig. 3). The sources clearly divide the continental and oceanic zones where the budget of *(E-P)i10* was calculated earlier. The criterion adopted here permits the investigation of the role of continental and oceanic moisture sources according to their location.

For the IRB in the WPR, the continental moisture sources were divide into East Africa (EA) extending to the west over the Sahel, the Arabian Peninsula (AP) mainly around the coast, in Asia to the west of this basin (West Asia, WA) and on the Indian

region (IR) (Fig. 4a). The oceanic moisture sources are easily divided and covers a small part in the western Mediterranean Sea (MEDT), the whole Red Sea (RS), the Persian Gulf (PG), the Indian Ocean mostly in the Arabian Sea region (IO), the Bay of Bengal (BB) and finally part of the Caspian Sea (CS). For the IRB in the WPR and MRP, the moisture sources almost remain in the same regions but change spatially because they are more extended in the MPR period with the exception of the IR and the BB, which are almost imperceptible (Fig. 4b). In the MPR the IO, which is extended to the south and southeast, is

highlighted. With respect to the rest of the continental sources, the EA is confined to the east of the African continent but the WA increased its spatial extent to the east and north. Because of the relative similar location of the sources for the GRB (Fig. 4c and d) and BRB (Fig. 4e and f), we kept the names already utilized for classification of the IRB moistures sources. However, some new region may appear such as a small moisture source to the north of the GRB, named Central Asia (CA), in the MPR. In the same period to the west of the BRB, we joined the areas previously classified for other basins as WA and IR into a single

region; this is because the position of the basin is totally located to the east of this region. Ordoñez et al. (2012) also divide the evaporative regions obtained in a backward analysis from the target region by taking into account the geographical well-known regions. Pathak et al. (2017) also calculated the moisture contribution from oceanic and terrestrial sources for the ISM rainfall. However, in their method they divided the core monsoon region into terrestrial sources that were approximately selected based on the uniform climate subtype of Köppen and the percentage of forest cover in the year 2000. while the oceanic sources

according to the vertically integrated moisture flux; divergent areas were considered the potential sources, whereas regions with high convergence were considered potential sink regions. In our approach and despite the consideration of all the monsoon core regions, the sources were classified after being identified as the most evaporative regions.

### 3.3 Role of continental and oceanic moisture sources

### 3.3.1 Budget of *(E-P)*

The budget of *(E-P)* along the 10 days cycles for the WPR and MPR and over the continental and oceanic regions and each basin separately was quantified (Fig. 5).

In the WPR, the *(E-P)i10* over the IRB itself is positive and greater than that obtained over the remaining continental and oceanic moisture sources (Fig. 5a). As seen in Figure 2, the Pet is greater that Pre over the IRB in these periods, which indicates





the prevalence of evaporative conditions in this basin. As the IRB is also a land-based source, the budget of *(E-P)i10* over the basin, together with the budget over the rest of the continental areas, reveals the importance of the continental moisture sources for the water supply to the IRB and probably because of the recycled moisture. Because the GRB and BRB occurs in both the continental and oceanic sources, the budget of the *(E-P)i10* remains positive (Fig. 5a). For the GRB, the positive *(E-P)i10* over

the continental sources is greater than previously obtained for the IRB and the BRB, but less than that obtained over the oceanic moisture sources of the IRB. Finally, the *(E-P)i10* over the BRB and its continental and oceanic sources are positive but less than previously computed over the moisture source regions of IRB and GRB. During the WPR in the GRB and the BRB, as occurred in the IRB, the Pet is greater than Pre and coinciding with evaporative conditions in the atmospheric column over them.

The budget of *(E-P)i10* was also obtained from the moisture sources delimited by p90 for the MPR (Fig. 5b) and when the precipitation increased over the three basins. In this period, as was previously discussed, the moisture sources are mostly larger and like those that occur in the Indian Ocean or West Asia (Fig. 3) and this could be reflected in the budget of *(E-P)i10*. Like in the WPR, the atmospheric moisture budget is positive but greater over continental than oceanic sources and the IRD itself, which confirms the results of Figure 2 showing that Pet is greater than Pre over this basin in the entire year. These results

indicate the increase of freshwater inputs to the basins due to continental evaporation (or recycling of moisture advected to the continents from remote regions). According to van der Ent et al. (2010), the continental evaporation recycling ratio is overall very high in Eurasia, which confirms that almost all of the continental evaporation returns to the continent, which can be seen from 50 to 100%, especially over China, which depends on its water resources almost entirely from terrestrial evaporation from the Eurasian continent. These findings confirm our results. The *(E-P)i10* values in the air masses tracked backward in

time from the GRB and BRB and reveal a negative budget over GRB and BRB itself (greatest for the BRB), which reflects that they act as an average moisture sink for humidity on air masses residing over them. In our approach, the resulting positive (negative) values of the moisture budget indicate moisture uptake $E > P$ (sinks $E < P$); however, as we do the interpretation of the water balance and not the single Evaporation or Precipitation, it could increase both $E$ and $P$ but one more than the other. Indeed, in this season, the precipitation exceeds the potential evapotranspiration in both the GRB and BRB (Fig. 2b and c). In

contrast, over the other terrestrial and oceanic sources of these basins, the budget is positive, which highlights the major amount of moisture uptake over the oceanic sources. Applying the Water Accounting Model described by van der Ent et al. (2010) and van der Ent and Savenije (2011), Nikoli et al. (2012) also found that among the nine global river basins studied at an annual scale, the Indus River basin shows the highest increase in evaporation, but due to the land-use change, the Ganges-Brahmaputra shows the highest precipitation increase (of continental origin).

### 3.3.2 Moisture contribution to precipitation (*(E-P) < 0*)

The percentage of moisture contribution from the IRB moisture sources is defined in Figure 5 and the IRB itself appears in Figure 6a. In both periods for the WPR, the MPR, and the IRB itself, the IO and WA are the most important. It can be appreciated in Figure 6a that the percentage of moisture supplied from continental sources represents a major percentage in





both periods under study. Although, in the MPR, the IO (38%) is the second most important source after the IRB itself (42%). To summarize these results, we calculated the seasonal average of $|(E-P)i10<0|$ from all of the continental and oceanic sources. To understand these averages it must be noted that basin's areas are not spatially of the same size; they were calculated at 1º in longitude and latitude.

The results confirm that terrestrial sources and overall the IRB itself can be responsible for the largest average moisture input to this basin (Fig. 6b). This result may seem erroneous because of the very well-known role of the Indian Ocean as a source of moisture for the Indian Monsoon. However, it must be understood that the moisture transported from the Indian Ocean contributes to precipitation processes throughout this region of Asia, and once it precipitates, it can evaporate and precipitate in the region and become the recycling that is fundamental to understanding this process. Karim and Veizer (2002) revealed

that evapotranspiration is the major route for the loss of water from the IRB. Besides, as river discharges fall short to reach the Sea during certain periods of the year, it is considered a closed basin (Molle et al., 2010). Thus, this increases the important role of evapotranspiration of natural vegetation and crops across the basin.

The same analysis performed in the air masses tracked backward in time from the GRB reveals that continental sources are the most important during the WPR for this basin, and among these, the utmost are the IR, GRB, and the WA (Fig. 6c). Among

the oceanic sources, the most important in this period are the BB and IO. In the MPR, the IO provides more than the 40% of the total atmospheric moisture influx, which is followed by the GRB itself (32%). The average moisture loss (contributing to precipitation) over the GRB in the WPR from continental sources is greater than oceanic (Fig. 6d) sources and the MPR; however, in both periods, the moisture contribution from the oceanic sources is greater than those occurring over the GRB in air masses residing over itself. Indeed, the GRB is responsible less than 1 mm/day of moisture loss over itself in the WPR. In

the MPR, the average contributions from all of the continental sources (including the GRB) is 12.8 mm/day, whereas from the oceanic sources, the contributions are less at approximately 11 mm/day. As the monsoon progresses in India, enhanced soil moisture and vegetation cover lead to increased evapotranspiration and recycled precipitation, which makes it possible for north-eastern India to have the highest recycling ratio (approximately 25%) (Pathak et al., 2014). Specifically, within the Ganges basin, the fraction of evaporation that ends up as precipitation is approximately 50-60% (Tuinenburg et al., 2012).

For the Brahmaputra basin, the most important moisture sources in the WPR are the IR and the BRB itself, BB and IO. In this period, the negative values in the atmospheric water balance over the basin, which were calculated over ten days in air masses tracked forward from them, represent 48% of the total moisture loss and this indicates that local moisture recycling must be favoured in this period (Fig. 6e). Indeed, it is shown in Figure 6e that continental sources are responsible for a major percentage of the moisture loss over the BRB in the WPR. Overall, for the MPR, the IO is the most important moisture source and is

responsible for the 37% of the total moisture loss over BRB. The BB is the second most important oceanic source while the rest of the oceanic sources are minimally important (even the CHS, which only appears in this season). The IR, GRB, and WA are among the continental moisture sources that are the most important in this period. An average of the total moisture loss over the BRB calculated as the contribution from oceanic and land-based moisture sources and including the BRB appears in Figure 6f. In the WPR, the major role of the continental regions as moisture sources for the BRB is clear, but in May-October,



the average $|(E-P)i10<0|$ is greater in air masses arriving at the basin from the oceanic sources (~22 mm/day). Nevertheless, there is not much difference from that computed in air masses with the continental origin.

We calculated the accumulated daily moisture contribution from the sources (from FLEXPART) and the precipitation over
them (from CHIRPS) along with the MPR, which is of utmost importance because of the monsoon influence in the 1981-2015 period. This analysis at a daily scale permits an understanding of the temporal relationship variability between the contribution of moisture from the sources to the precipitation (rapid increase & decrease) over the basins within the MPR, and this enables clarification, for example, in Figure 6b, the continental sources average supply to the GRB can be greater than from the oceanic sources.
For the IRB, the minimum rainfall accumulated anomalies occur on June 23 (Fig. 7a), and from this date onwards the rainfall is increased. At the beginning of June, the moisture supply to this basin was enhanced first by IO and later by WA and IRB itself. The anomalies on the contribution by the rest of the continental and oceanic sources occur after the abrupt rainfall increase over the basin and do not represent great changes to the amount of humidity according to low anomalies (Fig. 7a and b). Previous to the maximum accumulated anomaly of precipitation (on September 9), it is possible to observe a decay of
accumulated anomalies of $|(E-P)i10<0|$ values from the basin itself after the second half of August. A decrease of anomalies in the WA´s contribution starts less abruptly and a few days before the decay of the rainfall anomalies. From the beginning of the second half of August, the accumulated anomalies of moisture supply from the IO to the IRB starts to decrease; however, an abrupt decay is not clearly seen after it occurs for the precipitation.

Accumulated anomalies on the moisture contribution from the IO to the Ganges River Basin (GRB) during first days of June
reach the minimum value at  and then immediately increase rapidly (before the contribution from the rest of the sources); and later, on June 15, the minimum value of rainfall accumulated anomalies occurs over the basin (Fig. 7c). In fact, from the rest of oceanic sources, these values are positive during all of the MPR and do not surpass the 50 mm/day. These results show that the most significant amount of moisture to the GRB first comes from the IO, and the results of Figure 6d must be explained by the moisture recycling process over the continental sources of the GRB and /or a minor residence time of the water vapour
over the continent, which influences the budget of $(E-P)$. Among the continental sources, the accumulated anomalies of the contribution of moisture from the basin itself experiment at the beginning showed a similar cycle to the precipitation-accumulated anomalies, but later reaches the maximum value in the second half of August, days before September 16, when the rainfall actually reaches this point. From the rest of the continental sources, the annual cycle of accumulated anomalies reflects less of similarity than the rainfall. At the beginning of the second half of August, the accumulated anomalies from the
IO reach almost 500 mm/day, which confirms the huge amount of moisture transported from this source to the GRB. A day later, as previously commented, the precipitation anomaly falls and reflects a time response between moisture input to the basin from the IO and a rainfall decrease over it. However, a maximum value occurs from the BB when the anomalies for precipitation suggest the rainfall season is decaying. These results confirm that although the total moisture input to the GRB



during the MPR is greater from continental sources than from oceanic (Fig. 6d), the IO plays a crucial primary role on the hydrological cycle for the monsoonal precipitation onset over this basin.

Over the BRB, the seasonal accumulated anomaly of rainfall reaches a minimum on June 7 (Fig. 7e). However, before this date and around mid-May minimum values also occur in the accumulated anomalies of the moisture contribution from the BB and later at the end of May from the IO. After this, the moisture supply starts to increase from this region to precipitation over the BRB. Before the rainfall decay on September 13 (one day after maximum rainfall accumulated anomalies) decrease the moisture contribution occurs first from the IO and later from the BB towards the end of August. Both sources (as was discussed) are the main oceanic moisture sources for the BRB. From the continental sources of accumulated anomalies, the majority follow the accumulated anomaly of precipitation except for the moisture input from WAS, which is positive after the first days of May. Nevertheless, this region is not the most important continental source of moisture for the BRB.

Correlations were calculated between the total $|(E-P)i10<0|$ values computed from all of the sources and separately for the precipitation and potential evapotranspiration in the basins for the WPR and MPR. Significant $r$ values only appear in Figure 8. As expected, considering the annual cycle of the Pre and Pet at the basins, we obtained positive correlations between $|(E-P)i10<0|$ & Pre and negative correlations for $|(E-P)i10<0|$ & Pet. For the IRB in the WPR, the best positive correlations ($r >$ 0.60) are for the moisture input to the basin from the IO with precipitation, followed by significant r values also obtained with the contribution from the RS, PG, AP and the IRB itself. The moisture loss over the IRB is oppositely correlated with Pet in the basin, and only the moisture supply from the BB is not significantly correlated with the Pet. In the MPR, the only positive significant correlations were obtained for the precipitation and the moisture influx from EA, IRB itself, IR, and IO. For the monsoon season, no correlation was significant, which indicates that there is not a statistically direct relationship between Pet and $|(E-P)i10<0|$ during the MPR.

The correlations for the WPR and MPR in the GRB were express like that for the IRB and showed positive (negative) and statistically significant correlations for P & $|(E-P)i10<0|$ (Pet & $|(E-P)i10<0|$). The best positive feedback occurred for the series from the IO, BB, and IR and it was negative for the IO, EA, and AP. Some correlations are not significant, for example, the moisture contribution from the CHS, WA, CEA and the BB in the MPR. For the BRB, the analysis showed a contrast from the previous findings and few low and significant correlations for the moisture contribution from the BB, IR and the basin itself with the precipitation over this basin in the WPR. The Pet & $|(E-P)i10<0|$ correlations were negative for most of the cases in this period. In the monsoonal period, as seen in Figure 8, the r values are best correlated with both P and Pet with $|(E-P)i10<0|$ from the BB and IO, which are the two most important oceanic sources for the BRB. In addition, for the BRB, EA and the PG were only for the Pet.





A climatological analysis of the North American Monsoon System (NAMS) precipitation recycling reveals a positive feedback mechanism between monsoon precipitation and a subsequent increase in the precipitation of a recycled origin (Domínguez et al., 2008). For the wettest NAMS monsoons, Bosilovich et al. (2003) documented that the evaporation and soil wetness time series tends to track similarly to the precipitation. In the Gangetic Plain and north-eastern India, a significant amount of

precipitation also comes from precipitation recycling (Pathak et al., 2014). For example, for the GRB and at the initial phase of the monsoon, the Indian Ocean is a strong moisture source and the subsequent recharge of soil moisture makes the evapotranspiration over the Ganges basin become active after the onset of the monsoon (Pathak et al., 2017). Despite these results, we found negative correlations between the moisture contribution to the basins and the Pet on them; which suggests the need for a monthly analysis to determine whether or not it occurs at a minor temporal scale.

### 3.4 Variability of *(E-P)* during the ISM onset and the demise over the basins

We calculated the budget of *(E-P)i10* in air masses tracked backward in time from each basin at days -1, -4, -7 and -10 before the rainfall increase (decrease) associated with the South Asian Monsoon (SAM) onset (demise) over the IRB, GRB, and BRB. To determine the onset and demise dates, we applied an objective index from Noska and Misra (2016) for the basins, which

was based on the cumulative anomalies of averaged daily rainfall (see equations 4 and 5). To illustrate the method the Figure 9 shows the daily average precipitation over the GRB in 2010 and the cumulative anomalies. The cumulative anomalies reached the minimum value on June 15th and the maximum on September 22nd. For this year, the rainfall associated with the monsoon onset occurred on June 16 and ends on September 22. Observation indicates that the daily precipitation rate changes occur abruptly for the onset and demise, which agrees with similar findings for different regions across the Indian region and

Southeast Asia (e.g., Ananthakrishnan and Soman, 1988; Soman and Kumar, 1993; Cook and Buckley, 2009).

By applying equation 4 and 5, it was possible to obtain the onset and demise dates of precipitation associated with the monsoonal influence for every year. These dates are represented in Figure 10 and it is possible to observe that rainfall associated with the SAM onset starts first at the BRB (commonly in May), later the GRB (commonly in June) and finally at

the IRB (commonly in June, and some cases in July) (Fig. 10a). In contrast, the precipitation decline because of the SAM demise occurs first over the IRB, followed by the GRB and the BRB, which indicates that the length of the monsoonal rainy season at the IRB is shorter than over the GRB and both were shorter than over the BRB. This reveals that from the east to the west the onset of monsoon rainfall takes longer to occur. A climatology of the length of the summer monsoon season (in days) obtained by Misra and DiNapoli (2014) also reflects that over the region of the BRB the number of days between the onset

and demise is greater than in regions to the west (where the GRB and IRB are located) and where the length decreases longitudinally. Similar onset and retreat dates were obtained by Hasson et al. (2016) but utilizing a distinct method on a CMIP5 climate model's data for observational and future periods.



A composite of the days for the monsoonal rainfall onset and demise over each basin was performed. Utilizing each composite, the budget of (E-P) for days -1, -4, -7, and -10 was calculated before the onset and demise; this way, it was possible to determine the spatial changes of moisture uptake by the basins. One day backward in time from the onset at the IRB, air masses uptake humidity over the basin itself and the surrounding regions (Fig. 11). At day -4, air masses arriving at the IRB uptake humidity

from the western Indian Ocean, the Arabian Sea, the Persian Gulf, the continental regions to the west of the basin and the basin itself. Over the north-eastern Arabian Sea, a remarkable change from conditions of pre-monsoon onset days was also described by Howland and Sikdar (1983) when the specific humidity increased as much as 5 g/kg from pre-monsoon to monsoon onset. At days -7 and -10, the pattern of (E-P) is more extended with positive values (moisture uptake) mainly to the west of the basin, part of the Arabian Sea and the western Indian Ocean. Particles arrive at the IRB losing humidity from over the south

and south-east Asia and the Bay of Bengal.

Analysing the (E-P) pattern for the predemise, at day -1 it is very similar to the same day before the onset; however, in the centre and northeast of the basin appear (E-P) <0 areas, which indicates the prevalence of moisture loss. At day -4, areas with (E-P) >0 seem to occupy less than on day -4 of the preonset, whereas more parcels arrive at the basin after losing humidity (according to the greater spatial extension of areas of (E-P) <0). At days -7 and -10 of the predemise, the main differences on

the (E-P) pattern (with respect to the same days for the preonset) are over the basin, where greater (E-P) >0 values are apparent over the Arabian Sea at day -10 where moisture uptake is major for the predemise. This is because days before the demise there should be major precipitation and consequently greater moisture uptake for the basin.

One day backward in time from the monsoonal rainfall onset over the GRB, the air masses over this basin gain humidity almost over the entire basin itself, but to the east is a moisture sink, which in contrast covers practically the entire basin at day -1 from

the rainfall demise of this basin. At day -4 from the onset and on the budget of (E-P), the positive values are very intense for mainly those over the basin itself, to the west of the basin, over India, the Arabian Sea and part of the Bay of Bengal. For the day before the demise, the positive values in the field of (E-P) are restricted in the northern part of the Arabian Sea, which suggests this region plays a key role in the monsoonal rainfall onset but also the demise over the GRB. The negative values (moisture sink) are more intense to the east of the GRB before the demise (as expected). For the pattern of (E-P) at days -7 and

-10 from the onset, the WAS and IO plays as crucial role in providing humidity to this basin and their sources contribute for the same dates before the demise; however, the eastern part of the basin (on average) behaves as a moisture sink and (E-P)>0 values are more restricted to the north of the budget pattern.

For the GRB, the results of the backward experiment highlight that at day -1 from the onset this basin acts as a moisture sink for the region as a whole. In this day, the (E-P) reveals that air parcels arrive at the basin and gain humidity just from a small

region to the southwest of the basin. The pattern is very similar at day -1 from the demise but (E-P)>0 values are not located to the northeast of the basin. These results are not surprising since, from the Figure 2 results, we understand that over the MPR the water balance over the GRB suggests that precipitation exceeds evaporation. At day -4 of the onset, the basin uptake humidity from the west, the Indian region and the western part of the Bay of Bengal is visually noticeable. However, the (E-P) pattern completely changed for the day -4 from the demise, which shows that air masses arrive at the BRB and gain and





transport moisture from the west and north of the basin and from the small regions in the north-eastern Arabian Sea and the Bay of Bengal. Furthermore, moisture loss prevails in air masses traveling to the GRB from the south and when remaining over itself. At days -7 and -10, the spatial pattern of *(E-P)>0* is quite similar for the preonset and predemise with the most remarkable difference over the southeast of Asia, the Bay of Bengal, and the BRB itself due to the moisture loss prevalence.

### 3.5 Moisture contribution during the dry and wet conditions in the basins

The SPEI was utilized to identify dry and wet conditions at the IRB, GRB, and BRB. The temporal evolution of this index at the temporal scale of 6 months is shown in Figure 12. We identify dry conditions at the IRB from 1998 to 2002 and increasing wet conditions from 2011 to 2015. A drought intensive period in Pakistan was identified by Xie et al. (2013) for the late 1990s

10    to early 2000s in agreement with our results. Pakistan is mostly located within the IRB, and hence, the hydrological condition of the basin regulated those of the country. In the GRB during the 2000-2010 decade, drought conditions were very frequent, whereas in the BRB drought conditions occurred in 2003-2010 and 2012-2015 (Fig. 12). Kumar et al. (2013) documented that short-term drought (SPEI6) to the south of the basin over the Indian region is characterized by strong periodicity at quasi-biennial (2–4 years) and decadal (12–16 year) time scales.

We use the 6-month SPEI at the end of October (April) to diagnose dry and wet conditions at the basins over the MPR (WPR) seasons. We selected those seasons under severe and extremely dry and wet conditions according to SPEI6 values (Tables 1 and 2) and the anomalies on the moisture contribution ($|(E-P)i10<0|$) from each moisture source to the basins were calculated by creating composites of the WPR and MPR affected by severe and extremely dry and wet conditions. The SPEI6 was also

20    utilized for the same purposes by Drumond et al., (2016) to investigate drought episodes in the climatological sinks of the Mediterranean moisture source.

Common dry WPR occurred at the IRB and BRB in 2001 and at the GRB and the BRB in 1999. According to SPEI >1.5 values, severe and extreme wet WPR seasons occurred at the IRB in 2015 and 1983 (Table 1). In 2015, it was also severely wet in the GRB (as well as in 1982 and 1998), whereas at the BRB, just two seasons were classified as severely wet (2007 and

25    2010). In the period from 1981-2015, there were three severely dry MPR periods at the IRB and also three for the GRB but one of them was extremely dry (2014) (Table 2). For the BRB, despite being the wettest basin, four MPR are characterized under severely dry conditions and of these, the WPR of 2005 accounted for both the GRB and the BRB. During the wettest MPR periods (Table 2), the greatest number of cases occurred at the GRB as well, and all were severely wet (like the two periods in the BRB), whereas at the IRB, of the two wet periods in 2010 only one was extremely wet.

The moisture contribution negative anomalies for the WPR composites at the IRB are evidence of the major deficit in the moisture supply from the IR, IRB itself and the IO (Fig. 13a, orange bars); the same sources are responsible for greatest positive moisture loss anomalies for the wettest WPR (Fig. 13a, green bars). For the MPR, the anomalies in the moisture input to the IRB during dry periods occur mainly from two sources, the IO and the own IRB. This indicates that during the monsoonal





season under drought conditions, as it rains less over the basin it will not favour the precipitation over itself but could for remote regions. For the wettest MPR, the opposite occurs.

In the GRB, the driest WPR are associated with negative anomalies of moisture supply mainly from two continental moisture sources, IR and IO, and two oceanic sources, BB and IO (Fig. 13c, orange bars). The same sources are responsible for positive

anomalies during wettest WPR (green bars). This means that during the WPR months, dry and wet periods are regulated in the GRB by anomalies of the moisture supply from the surrounding land regions (mainly to the south over India) and the Bay of Bengal and less from the IO. For MPR, the greatest negative anomalies of the $|(E-P)i10<0|$ values over the GRB in the composite of the dry conditions occur in air masses arriving at the basin from itself, the IO and WA, whereas for wettest periods the highest positive anomalies are on the moisture inputs to the GRB from the IO and followed by the GRB itself and

the WA. These anomalies allow confirmation that the wettest periods in the GRB are related to an increase of the moisture supply from the IO and the local contribution is surely enhanced because of moisture recycling, which is a mechanism well explained for the GRB by Tuinenburg et al. (2012).

In the BRB during the WPR as well as for the GRB, the IR, BB, IO and the basin itself are the regions from where a reduction of moisture supply to the BRB drastically occurs during the driest November-April periods (Fig. 13e, orange bars) and the

moisture supply increases during the wettest periods (green bars). In the MPR, the IO becomes the source from which the atmospheric transport that contributes to precipitation over the BRB experiment shows the highest reduction during the driest periods (< 3 mm/day) (Fig. 13f, orange bar) and wettest periods (> 6 mm/day) (Fig. 13f, green bar). The BB is the second most important oceanic source in terms of the anomalies, whereas the IR is the most important among the terrestrial sources. For almost all of the cases when dry/wet conditions occur at the basins, negative/positive anomalies occur for the moisture

contribution to precipitation, which is generally from all of the sources over the basins. As precipitation depends on the amount of water vapour, the most important anomalies in the contribution from these sources highlight the main aspects responsible for drought and intense precipitation over the basins.

## 4 Conclusions

The 3-dimensional model FLEXPART was used to track backward in time the air masses residing over the Indus, Ganges, and Brahmaputra River basins (IRB, GRB, and BRB). The model permitted the calculation of the budget of evaporation minus precipitation *(E-P)* along backward and forward trajectories integrated over ten days and allowed the identification of the climatological moisture sources of each basin for the Westerly Precipitation Regime (WPR) (November-April) and Monsoonal Precipitation Regime (MPR) (May-October) over 35 years (1981-2015). The results indicate that moisture sources are

positioned in continental and oceanic regions as well as the basins themselves. Their spatial extension increases during the MPR (when the rainfall is highest) and principally in the Indian Ocean. Along each trajectory, the budget of *(E-P)* over most evaporative continental and oceanic sources was calculated, which revealed the importance of moisture uptake for the basins over continental regions during the WPR. A forward analysis performed from the sources revealed the important role of continental regions on the average moisture contribution to precipitation over the IRB and GRB during the MPR and during





which the oceanic sources are the most important for the BRB. However, during the MPR, the greatest moisture contribution to precipitation over the basins occurs from the IO, except for the IRB, where local moisture losses in the (E-P) play a dominant role. Additionally, the IO seems to be responsible for first providing moisture to the basins in the MPR period and is linked to the rapid rainfall increase/decrease. Generally, the most important moisture sources for the IRB, GRB and BRB are the west

Asia extension (WAS), the Indian region (IR), the Indian Ocean (IO), the Bay of Bengal (BB) and the basins itself. A spatial analysis of the resulting *(E-P)* pattern in the preonset and predemise of the monsoonal precipitation over each basin, exposed the spatial differences mainly on the moisture uptake variability and confirmed the spatial reduction mainly of the evaporative source in the IO days before the demise.

As expected, the average moisture (summed *(E-P)<0* from all the sources) loss over the basins' values integrated over ten days

is positively correlated with the precipitation and negatively correlated with the potential evapotranspiration even during the MPR, when some studies suggest that both variables increase. The roles of the sources in the moisture contribution to precipitation during severe and extremely dry and wet conditions at the basins were assessed through WPR and MPR composites, and confirmed the crucial role of those most important moisture sources (eg. IR, IO and the basins itself) in providing less (more) humidity during dry (wet) conditions in both periods WPR and MPR. Even though the hydrological

cycle over the Asian region has been widely investigated, the results obtained here will also support further challenges to research variability and extreme phenomena but specifically over the IRB, GRB and BRB. Future research would be an important contribution to investigating the influence of the Modes of Climate Variability, principally ENSO, on the modulation of moisture transport from the sources of moisture of the basins.

**Acknowledgements.** This work was supported by EPhysLab (UVIGO-CSIC Associated Unit). R. Sorí would like to acknowledge the grant received by the Xunta of Galicia, Spain, in support of his doctoral research work; R. Nieto acknowledges the support provided by CNPq grant 314734/2014-7 from the Brazilian government; A. Drumond acknowledges the support of the Spanish Government and FEDER via the SETH (CGL2014-60849-JIN) project. Thanks to the IMDROFLOOD project financed by the Water Works 2014 co-funded call of the European Commission.

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

**Table 1.** WPR under severe and extremely dry and wet conditions at the Indus, Ganges and Brahmaputra River basins during
the period from 1981 – 2015.

| month/year Dry | IRB SPEI-6 | month/year | GRB SPEI-6 | month/year | BRB SPEI-6 |
|---|---|---|---|---|---|
| 04/2001 | -1.64 | 04/1999 | -1.55 | 04/2001 | -1.72 |
|  |  | 04/2009 | -2.25 | 04/2014 | -1.88 |
|  |  |  |  | 04/1999 | -1.88 |
| *Wet* |  |  |  |  |  |
| 04/2015 | 1.51 | 04/1982 | 1.78 | 04/2007 | 1.69 |
| 04/1983 | 2.0 | 04/2015 | 1.89 | 04/2010 | 1.92 |
|  |  | 04/1998 | 2.0 |  |  |

**Table 2.** MPR under severe and extremely dry and wet conditions at the Indus, Ganges and Brahmaputra River basins during
the period from 1981 – 2015.

| month/year Dry | IRB SPEI-6 | month/year | GRB SPEI-6 | month/year | BRB SPEI-6 |
|---|---|---|---|---|---|
| 10/1991 | -1.51 | 10/2005 | -1.56 | 10/1994 | -1.58 |
| 10/1987 | -1.62 | 10/1992 | -1.68 | 10/2006 | -1.60 |
| 10/2009 | -1.74 | 10/2014 | -2.35 | 10/2005 | -1.60 |
|  |  |  |  | 10/1982 | -1.61 |
| *Wet* |  |  |  |  |  |
| 10/2015 | 1.75 | 10/1999 | 1.62 | 10/1998 | 1.56 |
| 10/2010 | 2.08 | 10/2013 | 1.65 | 10/1988 | 1.72 |
|  |  | 10/2011 | 1.83 |  |  |
|  |  | 10/1990 | 1.92 |  |  |





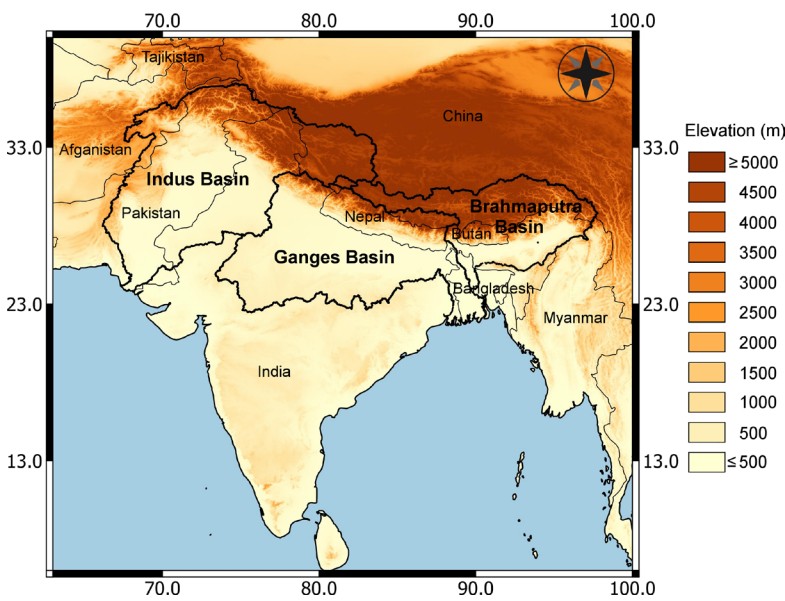

**Figure 1:** The geographic location and boundaries of the Indus, Ganges and Brahmaputra river basins from Hasson et al. (2013); and the elevation from the Hydrosheds project (Lehner et al., 2008).

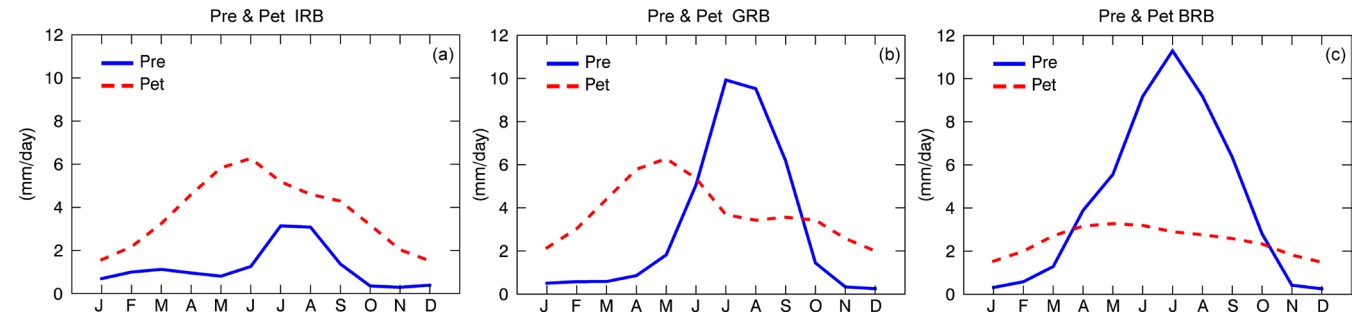

**Figure 2.** The 1981-2015 annual cycle of precipitation (blue lines, mm/day) and potential evapotranspiration (discontinued red lines mm/day) over the Indus (a), Ganges (b), and Brahmaputra (c) river basins.



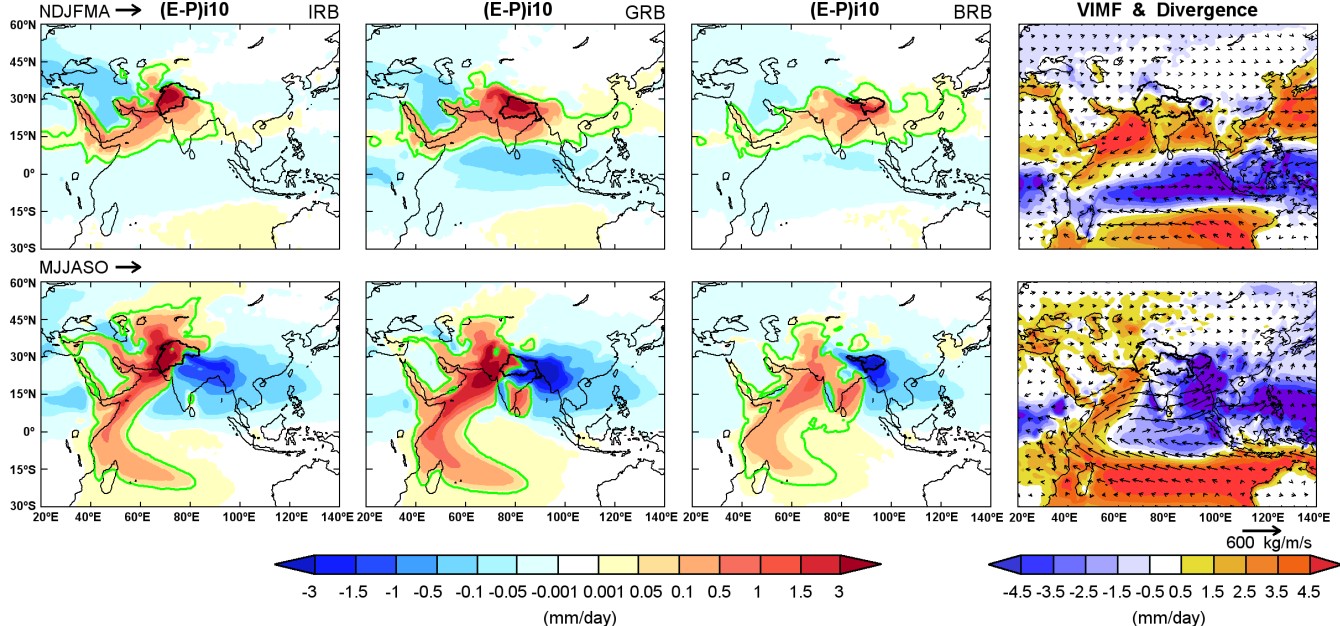

**Figure 3.** November-April (top) and May-October (bottom) *(E-P)i10* (mm/day) backward integrated from the Indus, Ganges and Brahmaputra river basins (contoured by a black line), Vertically Integrated Moisture Flux (VIMF) (arrows, kg/m.s) and Divergence of the VIMF (shaded, mm/day). The 90[th] percentile is represented by a green line. The period of study was 1981-2015.





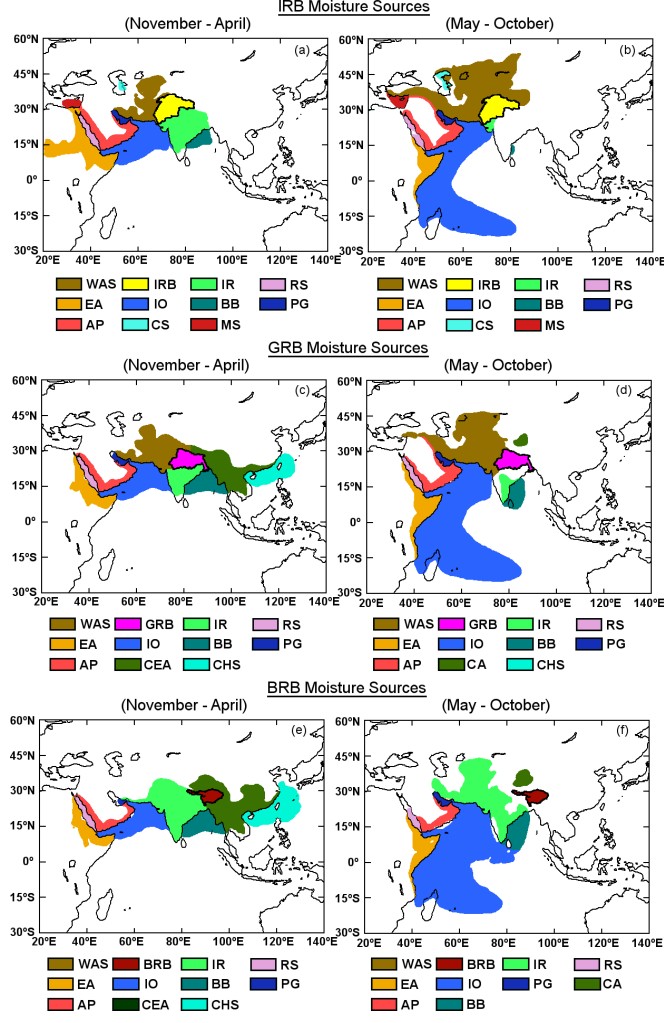

**Figure 4.** Schematic representation of the IRB (a and b), GRB (c and d) and BRB (e and f) moisture sources delimited by the p90 value shown in Figure 3 for the WPR (left column) and MPR (right column). The acronyms identifying the moisture sources are defined in the text.





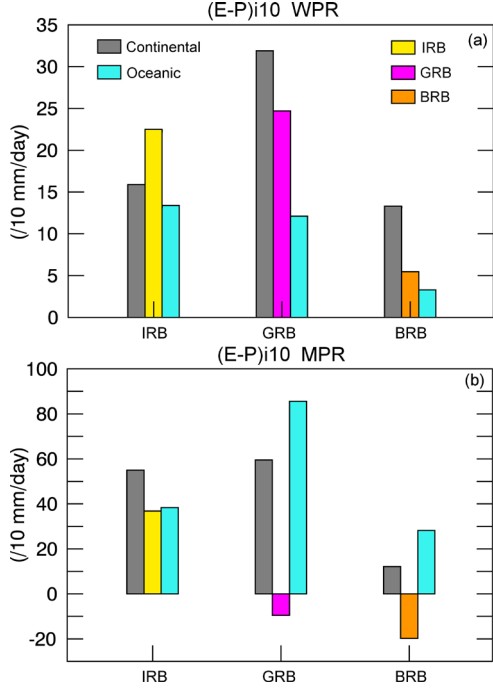

**Figure 5.** Total budget of *(E-P)* integrated over 10 days in air masses tracked backward in time from the basins, over continental sources, oceanic sources and the basins themselves. For the WPR (November-April) and the MPR (May-October) in the period from 1981-2015.





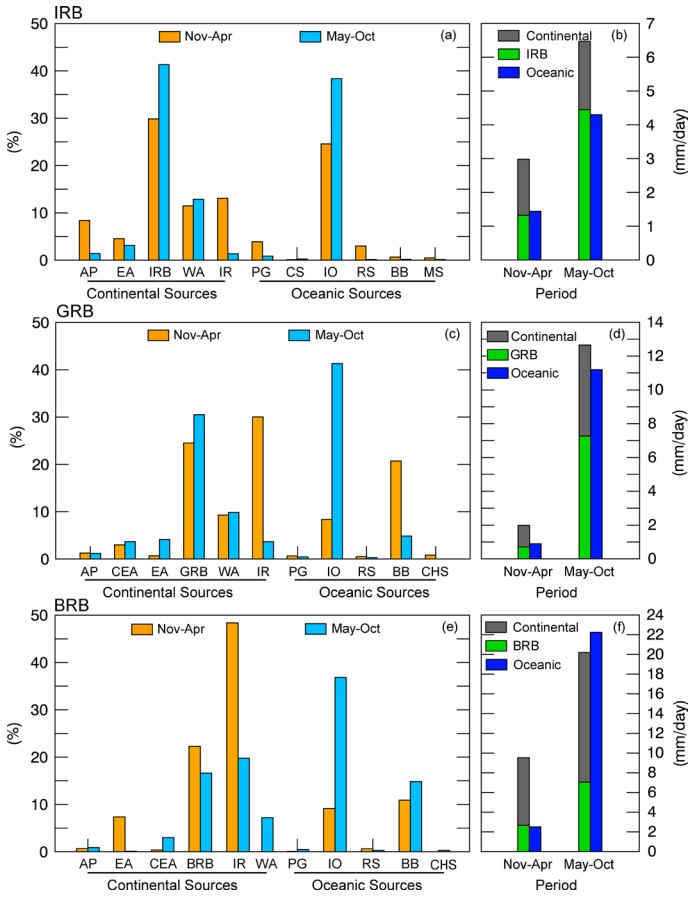

**Figure 6**. (left column) The percentage of moisture contributions ($|(E-P)i10<0|$) from the moisture sources to the IRB (a), GRB (c) and BRB (e) during November-April (WPR) (orange bars) and May-October (MPR) (blue bars). (right column) The average moisture contribution from continental sources (grey bars), the IRB (b), GRB (d), BRB (f) (green bars) and oceanic sources (dark blue bars).





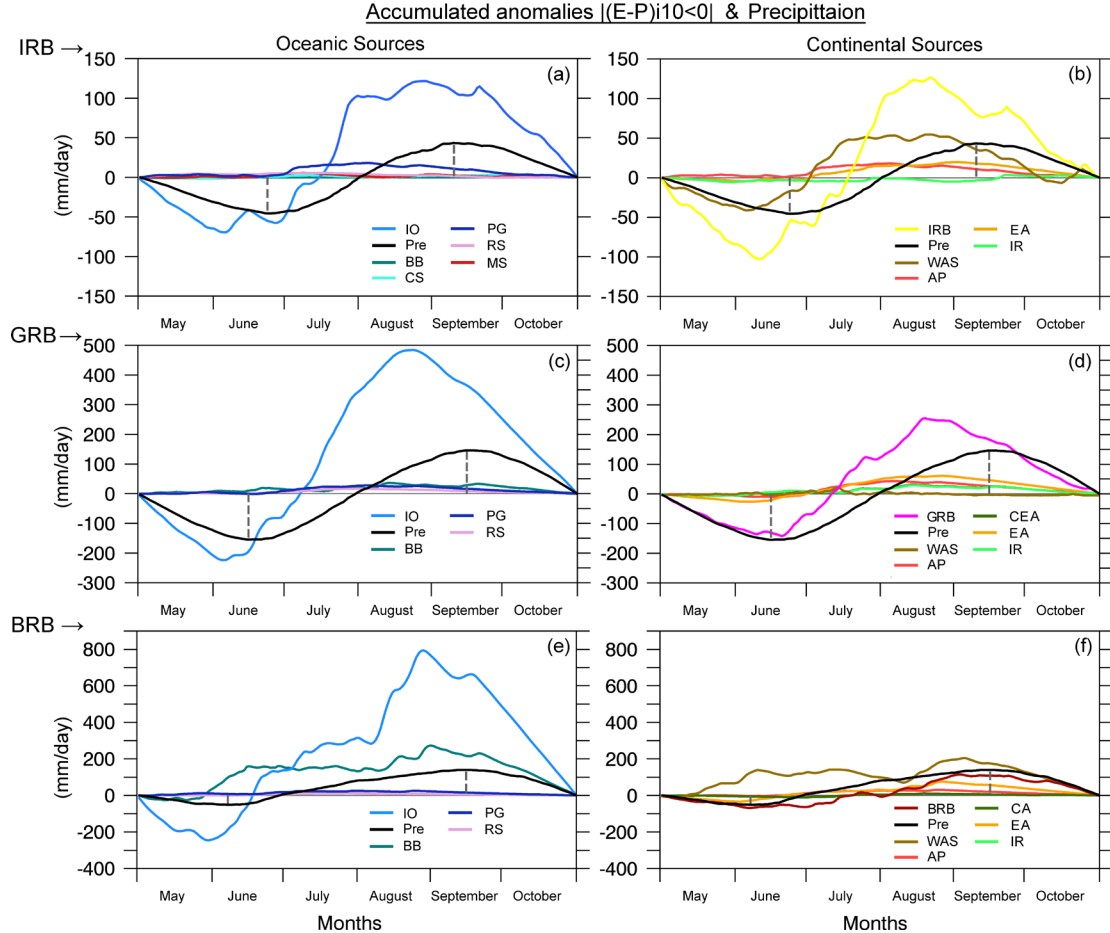

**Figure 7.** Accumulated anomalies of *|(E-P)i10<0|* values computed over each basin on air masses forward in time and tracked from the oceanic sources (left panel), continental sources (right panel) (the colour of the lines are in accordance with the name of the sources in Fig. 5) and precipitation (red line).





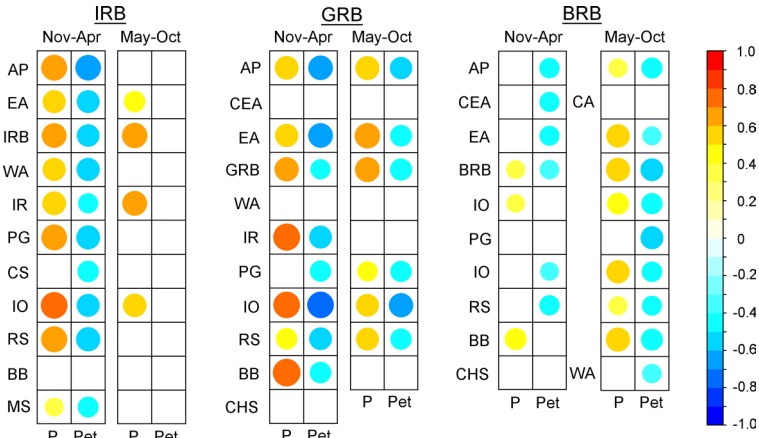

**Figure 8.** Monthly correlations for the WPR (Nov-Apr) and MPR (May-Oct) periods between precipitation and potential evapotranspiration (*Pre, Pet*; from CRU) with total (summed average contributions from all the sources) $|(E-P)i10<0|$ over each basin (from FLEXPART).

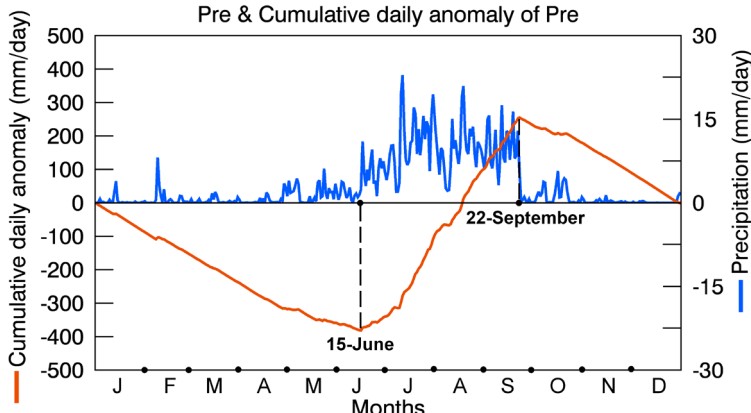

**Figure 9.** Daily precipitation (blue line) and the cumulative daily anomaly of the precipitation (orange line) over the GRB during 2010. June 15 (September 22) represents the minimum (maximum) cumulative daily anomaly of the precipitation.





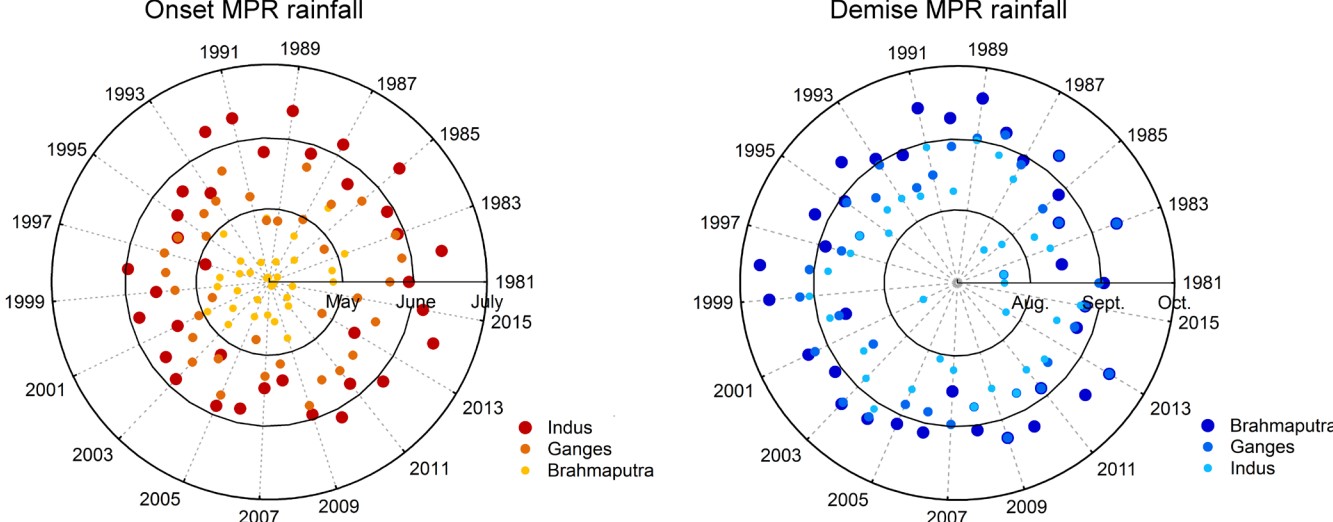

**Figure 10.** Onset and demise of the monsoonal rainfall for the Indus, Ganges, and Brahmaputra river basins.



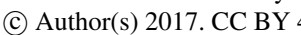

**Figure 11.** Composite of *(E-P)* in a backward experiment from the IRB for a composite of days -1, -4, -7 and -10 from the Onset and Demise of the monsoon.





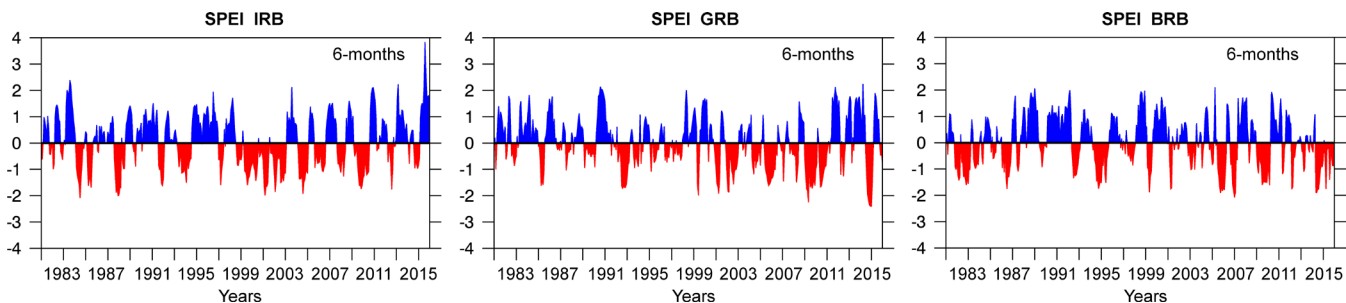

**Figure 12.** Monthly SPEI for a time scale of 6 months averaged for the Indus, the Ganges, and Brahmaputra River basins in the period from 1981 – 2015.

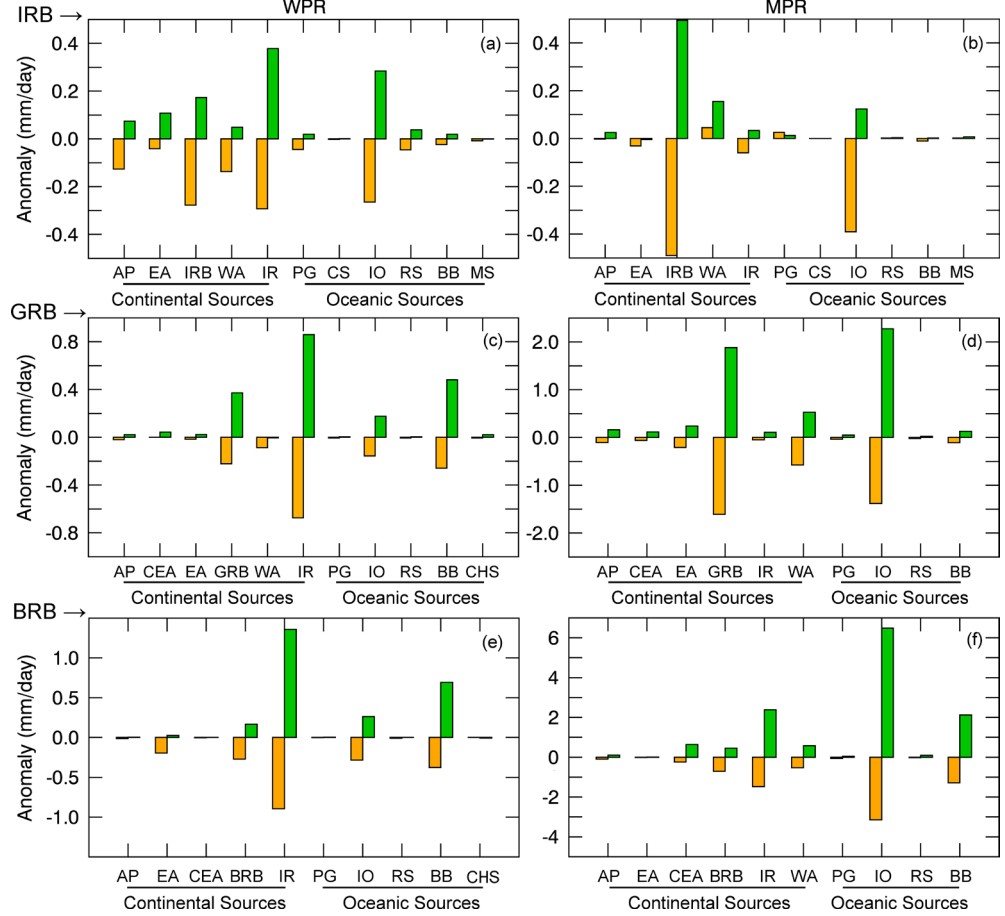

**Figure 13.** Anomalies of the moisture contribution ($|(E-P)i10<0|$) from each source to the IRB, GRB, and BRB during severe and extremely dry and wet condition at the basins (orange and green bars, respectively) from the period of 1981 – 2015.

