# Peer review of "The atmospheric branch of the hydrological cycle over the Indus, Ganges and Brahmaputra River basins"

_Hydrology and Earth System Sciences, 2017_

## Referee Comment (RC1) · Anonymous Referee #1 · 15 Aug 2017

Review of: The atmospheric branch of the hydrological cycle over the Indus, Ganges and Brahmaputra River basins
Authors: Rogert Sori et al.
Recommendation: Accept after some revisions

Review:
The authors have conducted a detailed water budget of the three major river basins of the Asian continent. Their results of the dominance of local recycling of moisture within the river basins followed by the Indian Ocean and rest of the continental sources is quite interesting. Overall the paper is well written and the figures are of good quality. Some of the most substantive suggestions are as follows:

1) The authors should discuss the quality of ERA-Interim analysis for the region especially in relation to the 4D structure of moisture, which is critical to the conclusions drawn in the paper.

2) Will the inconsistency of using E and P and the other meteorological variables from independently different sources have some bearing on the result? Does E-P from ERA-Interim look like Fig. 2? If not then how much of that is affecting the results of the paper?

3) Is there any merit in using E-P from ERA-Interim even if it does validate with other independent observations because of balancing the water budget?

---

## Referee Comment (RC2) · Anonymous Referee #2 · 16 Aug 2017

The authors have performed a very important analysis on the moisture transport that controls the water budget of IGB. The topic is interesting and suits well to the scope of the journal. However, I have few comments, which the authors may address.

1. The authors have mentioned about Pathak et al. (2017). It is not clear, what are the extra added information/ findings from this analysis with respect to Pathak et al. (2017). I can see that majority of the conslusions are similar to both Pathak et al (2017) and Pathak et al (2015). It would be good, if the authors highlight in details of the agreements and disagreements with Pathak et al (2015) and Pathak et al. (2017) and specify the added findings from the present analysis.

[Figure]

2. The value of P-E in a water budget equation for monthly scales typically equates to the divergence with a negative sign. Often it is observed that such divergence is a better estimate when we perform any analysis at a monthly scale. Further to this, a recent article shows that uncertainty across reanalysis can be reduced by considering divergence. I can understand that the authors probably need daily data for their analysis, but a comment or discussion with some scope for future research may be a good addition. This is specifically because, the authors have used a single reanalysis, and use of multiple reanalysis may increase uncertainty, which we need to reduce. The authors may follow: https://www.nature.com/articles/srep29664

3. The authors have used CRU data and I am just wondering if they will get similar conclusions with IMD/ Aphrodite data. As the IMD/ Aphrodite uses more number of stations, and hence, such a check is better to be performed. Just a suggestion from this reviewer.

4. I have some concern about combining Arabian sea and Indian Ocean. I would rather be interested in considering them seprately. This will give us some idea of the relative contributions from them.

5. Some comments on the contribution from Monsoon Depressions and their role in water cycle would be of great interest.

---

## Author Comment (AC1) · 28 Aug 2017

**Dear Reviewer 1.**

Thank you for your time and the suggestions to improve our paper.

Please find below the reply for all of your comments.

1) The authors should discuss the quality of ERA-Interim analysis for the region especially in relation to the 4D structure of moisture, which is critical to the conclusions drawn in the paper.

R-1. Thank you. Please consider the paragraph bellow, which was added to the paper. It has been difficult to find a research concerning the ERA-Interim 4D improvements impacts in the atmospheric moisture field structure, specifically for the region under study.

The use in ERA-Interim of 4D-Var data assimilation contributed to better time consistency than the 3D-Var used in ERA-40. However, the agreement between the global tendencies of mass and total column water vapour (TCWV) and (E − P) is not very good in ERA-Interim, but it is still much better than for ERA-40 where (E − P) (Berrisford, 2011).

Sebastian et al. (2016) found a huge uncertainty in the estimates of (P–E) over South Asia, when computed from different reanalysis, but recommend to use atmospheric budget for computation of water availability in terms of (P–E) rather than based on individual values of P and E. We also consider that in the state of the art discussion of three reanalysis (ERA-I, MERRA and CFRS), Lorenz and Kunstmann (2012) obtained that the ERA-Interim shows both a comparatively reasonable closure of the terrestrial and atmospheric water balance and a reasonable agreement with the observation datasets. This findings support the use of ERA-Interim reanalysis for running FLEXPART in order to reduce the uncertainty in this study. In the same way, the Vertical Integrated northward and eastward Moisture Flux data to calculate the Vertical Integrated Moisture Flux (VIMF) and it´s Divergence belong to the ERA-Interim Reanalysis with a resolution of 1°×1°. Computing the (P–E) directly from atmospheric budget with divergence of moisture flux for different reanalyses improved correlation with observed values of (P–E) according to Sebastian et al. (2016) results; what we consider to do in future studies to evaluate the ability of different reanalysis in the representation of the moisture budget for the target region.

2) Will the inconsistency of using E and P and the other meteorological variables from independently different sources have some bearing on the result? Does E-P from ERA-Interim look like Fig. 2? If not then how much of that is affecting the results of the paper?

R-2. Thank you for your questions. The idea of using datasets from different sources was first because certain datasets like from CRU permitted to calculate easily the SPEI. Daily data from CHIRPS also made possible to utilize a combination of rain gauge data and satellite to stablish onset and demise dates for the monsoon in the basins. An advantage of CHIPRS is that the data cover all the period under study while from Aphrodite are only available until 2007. We made a new Figure 2, described the results, and compared them with previous findings (Please read the section 3.1). The next sentences have been also added to the paper.

3.1 The precipitation and evaporation over the basins

The mean annual cycle of the Precipitation (P), Evaporation (E) and Potential evapotranspiration (Pet) over the Indus, Ganges and Brahmaputra basins appears in Figure 2. For the three basins, the maximum P occurs during the summer months. It can be observed that monthly P values from ERA-I tend to be slightly greater than those computed from CRU, but the annual cycle is the same. These differences are best appreciated in the annual cycle of P over the BRB. In the IRB, the P annual cycle is characterized by two maximum peaks in February-March and July-August (Fig. 2a). The E follows approximately this cycle but with lower values. In this basin, the Pet remains higher than the P and E across the year; in fact, Cheema (2012) argue

that the major part of this basin is dry and located in arid to semiarid climatic zones. Laghari et al. (2012) also found for the climatology from 1950–2000, that Pet exceeds P at the IRB across the year. Pet is enhanced after maximum precipitation; maximum values occur in May-June. Over the GRB maximum P occurs between May and October and is greater than over the IRB. The Pet and E annual cycles over this basin differ, and as expected, Pet > E. The Pet annual cycle is mainly like for the IRB. Indeed, both variables reflect close but different information. The E annual cycle agrees to that obtained by Hasson et al. (2014) for the three river basins. Over the BRB, the monthly average precipitation both from CRU and ERA-I increases abruptly from March until a maximum (>11.0 mm/day) in July and later falls until a minimum is reached in December (Fig. 2c). The Pet and E are very close and does not surpass 4 mm/day in the annual climatology. Particularly the Pet annual cycle highlights for being lower than what was obtained for the IRB and GRB. The annual cycle of P (from CRU and ERA-I) and E for the IRB, GRB and BRB follow the same annual cycle than those obtained by Hasson et al., (2014). These authors analysed the seasonality of the hydrological cycle over the same basins for the 20th century climate (1961–2000 period) utilizing PCMDI/CMIP3 general circulation models (GCMs) and observed precipitation data.

Tropical cyclones and weak disturbances contribute to monsoon rainfall. Among these systems, the most efficient rain-producing system (responsible of about half of the Indian summer monsoon rainfall) is known as the Indian monsoon depression (MD) which generally forms around Bay of Bengal and propagates westward or northwestward with the typical life span of three to six days (Ramage 1971; Yoon and Huang, 2012). The change in the large-scale circulation, especially the converging atmospheric water vapour flux is responsible for the MD modulation by the 30-60 day monsoon mode (Yoon and Huang, 2012). Over the Brahmaputra basin, the rainiest, heavy rainstorms are due to the shifting of the eastern end of the seasonal monsoon trough to the foothills of Himalayas in the north and the 'Break' monsoon situations during the monsoon season (Dhar and Nandargi, 2000). Summarizing, the BRB is wetter than the western GRB and IRB; this is because the monsoon rainfall dominates in the summer months in the eastern region and gets weaker on the western side with a time delay of a period of weeks (Hasson et al., 2014).

[Figure]

3) Is there any merit in using E-P from ERA -Interim even if it does validate with other independent observations because of balancing the water budget?

R-3. Please consider this sentences we already included in the paper.

Sebastian et al. (2016) found a huge uncertainty in the estimates of (P−E) over South Asia, when computed from different reanalysis, but recommend to use atmospheric budget for computation of water availability in terms of P−E rather than based on individual values of P and E. We also consider that in the state of the art discussion of three reanalysis (ERA-I, MERRA and CFRS), Lorenz and Kunstmann (2012) obtained that the ERA-Interim shows both a comparatively reasonable closure of the terrestrial and atmospheric water balance and a reasonable agreement with the observation datasets. This findings support the use of ERA-Interim reanalysis for running FLEXPART in order to reduce the uncertainty in this study. In the same way, the Vertical Integrated northward and eastward Moisture Flux data to calculate the Vertical Integrated Moisture Flux (VIMF) and it´s Divergence belong to the ERA-Interim Reanalysis with a resolution of 1° × 1°. Computing the P−E directly from atmospheric budget with divergence of moisture flux for different reanalyses improved correlation with observed values of P−E according to Sebastian et al. (2016) results; What we consider to do in future studies to evaluate the ability of different reanalysis in the representation of the moisture budget for the target region.

---

## Author Comment (AC2) · 28 Aug 2017

**Dear Reviewer 2.**

Thank you for your time and the suggestions to improve our paper.

Please find below the reply to your comments.

1. The authors have mentioned about Pathak et al. (2017). It is not clear, what are the extra added information/ findings from this analysis with respect to Pathak et al. (2017). I can see that majority of the conslusions are similar to both Pathak et al (2017) and Pathak et al (2015). It would be good, if the authors highlight in details of the agreements and disagreements with Pathak et al (2015) and Pathak et al. (2017) and specify the added findings from the present analysis.

R-1. Thank you for your suggestion, you are right. You can find in the new version new paragraphs concerning the main agreements and disagreements with Pathak et al (2014) and Pathak et al. (2017).

Pathak et al. (2017) also calculated the moisture contribution from oceanic and terrestrial sources for the ISM rainfall. However, in their method, the terrestrial sources were approximately selected based on the uniform climate subtype of Köppen and the percentage of forest cover in the year 2000; while the oceanic sources according to the VIMF. They considered divergent areas as the potential sources, whereas regions with high convergence were considered potential sink regions. Nevertheless, in our approach, moisture sources are considered those regions from where air masses uptake humidity before arrive to the basins.

Gimeno et al. (2010) obtained that the Red Sea source provides vast amounts of moisture that precipitate between the Gulf of Guinea and Indochina in June-August. Besides, Pathack et al. (2017) noted that a significant fraction of atmospheric moisture to the ISMR comes from five main moisture sources: the western Indian Ocean, central Indian Ocean, upper Indian Ocean, Ganges basin, and Red Sea and the neighbouring gulf. In agreement with the previous findings, we obtained that the Red Sea and the Persian Gulf acts as sources of moisture for the Indus, Ganges and Brahmaputra River basins. Nevertheless, in our analysis we considered them separated (not like Pathack et al., 2017), obtaining a negligible role from each one to the total moisture contribution mainly for the GRB and BRB in both the WPR and MPR.

It may be confusing that the total contribution to precipitation from continental sources is a little greater than from ocean sources for the IRB and GRB in the MPR (Fig. 6b,d); contrary of Pathack et al. (2017) results for the Indian region, which mostly comprises the GRB. Differences may arise because the method to calculate the moisture contribution, both based in a Lagrangian approach. Particularly Pathack et al. (2017) implemented an extension of the Dynamic Recycling Model (DRM) developed by Dominguez et al. (2006) and modified by Martinez and Dominguez (2014). Their method permits to quantify the relative contributions from different sources to the atmospheric moisture over a given sink region by calculating the fraction of atmospheric moisture collected by an air column along its trajectory between times considering the evaporation and the precipitable water, respectively, along the two-dimensional trajectory.

These results confirm that although the total moisture input to the GRB during the MPR is greater from continental sources than from oceanic (Fig. 6d), the IO plays a crucial primary role on the hydrological cycle for the monsoonal precipitation onset over this basin; agreeing to Pathack et al (2014; 2017), who highlight the key role of the IO on the Indian summer monsoon and the land surface processes role in the generation of precipitation within the Indian subcontinent.

2. The value of P-E in a water budget equation for monthly scales typically equates to the divergence with a negative sign. Often it is observed that such divergence is a better estimate when we perform any analysis at a monthly scale. Further to this, a recent article shows that uncertainty across reanalysis can be reduced by considering divergence. I can understand that the authors probably need daily data for their analysis, but a comment or discussion with some scope for future research may be a good addition. This is specifically because, the authors have used a single reanalysis, and use of multiple reanalysis may increase uncertainty, which we need to reduce. The authors may follow: https://www.nature.com/articles/srep29664.

R-2. Thank you. We tried to improve this explanation according to your suggestions:

Sebastian et al. (2016) found a huge uncertainty in the estimates of (P–E) over South Asia, when computed from different reanalysis, but recommend to use atmospheric budget for computation of water availability in terms of P–E rather than based on individual values of P and E. We also consider that in the state of the art discussion of three reanalysis (ERA-I, MERRA and CFRS), Lorenz and Kunstmann (2012) obtained that the ERA-Interim shows both a comparatively reasonable closure of the terrestrial and atmospheric water balance and a reasonable agreement with the observation datasets. This findings support the use of ERA-Interim reanalysis for running FLEXPART in order to reduce the uncertainty in this study. In the same way, the Vertical Integrated northward and eastward Moisture Flux data to calculate the Vertical Integrated Moisture Flux (VIMF) and it´s Divergence belong to the ERA-Interim Reanalysis with a resolution of $1° \times 1°$. Computing the P–E directly from atmospheric budget with divergence of moisture flux for different reanalyses improved correlation with observed values of P–E according to Sebastian et al. (2016) results; what we consider to do in future studies to evaluate the ability of different reanalysis in the representation of the moisture budget for the target region.

3. The authors have used CRU data and I am just wondering if they will get similar conclusions with IMD/ Aphrodite data. As the IMD/ Aphrodite uses more number of stations, and hence, such a check is better to be performed. Just a suggestion from this reviewer.

R-3. Thank you for your suggestion. Yes, for sure the quality of observational data from Aphrodite is an advantage. However, Aphrodite does not have AED data, which are necessary to estimate the weight of the water demand by the atmosphere in drought conditions through the SPEI, and also their availability is just until 2007, while our study covers until 2015.

4. I have some concern about combining Arabian Sea and Indian Ocean. I would rather be interested in considering them separately. This will give us some idea of the relative contributions from them.

R-4. Thank you. We understand your concern. However, please take into account that to do it we should implement another method to delimitate the boundaries of the sources, perhaps based on geographical borders. That would mean having to track backward and forward in time air masses from different sources and calculate once more the (E-P) along the trajectories. A further work could be done to evaluate specifically the role of the Arabian Sea.

5. Some comments on the contribution from Monsoon Depressions and their role in water cycle would be of great interest.

R-5. Thank you for this important suggestion. We added in the article the paragraph below.

Tropical cyclones and weak disturbances contribute to monsoon rainfall. Among these systems, the most efficient rain-producing system (responsible of about half of the Indian summer monsoon rainfall) is known as the Indian monsoon depression (MD) which generally forms around Bay of Bengal and propagates westward or northwestward with the typical life span of three to six days (Ramage 1971; Yoon and Huang, 2012). The change in the large-scale circulation, especially the converging atmospheric water vapour flux is responsible for modulation of a MD by the 30-60 day monsoon mode (Yoon and Huang, 2012). Over the Brahmaputra basin, the rainiest, heavy rainstorms are due to the shifting of the eastern end of the seasonal monsoon trough to the foothills of Himalayas in the north and the 'Break' monsoon situations during the monsoon season (Dhar and Nandargi, 2000). Summarizing, the BRB is wetter than the western GRB and IRB; this is because the monsoon rainfall dominates in the summer months in the eastern region and gets weaker on the western side with a time delay of a period of weeks (Hasson et al., 2014).